# Design and Experiment of Automatic Transport System for Planting Plate in Plant Factory

**Dongdong Jia** [1,2], **Wenzhong Guo** [2], **Lichun Wang** [2], **Wengang Zheng** [2,*] **and Guohua Gao** [1,*]

1 College of Mechanical & Energy Engineering, Beijing University of Technology, Beijing 100124, China; jiadd@nercita.org.cn
2 Research Center of Intelligent Equipment, Beijing Academy of Agriculture and Forestry Sciences, Beijing 100097, China; guowz@nercita.org.cn (W.G.); wanglc@nercita.org.cn (L.W.)
* Correspondence: zhengwg@nercita.org.cn (W.Z.); gaoguohua@bjut.edu.cn (G.G.)

**Abstract:** In the plant factories using stereoscopic cultivation systems, the cultivation plate transport equipment is an essential component of production. However, there are problems, such as high labor intensity, low levels of automation, and poor versatility of existing solutions, that can affect the efficiency of cultivation plate transport processes. To address these issues, this study designed a cultivation plate transport system that can automatically input and output cultivation plates, and can flexibly adjust its structure to accommodate different cultivation frame heights. We elucidated the working principles of the transport system and carried out structural design and parameter calculation for the lift cart, input actuator, and output actuator. In the input process, we used dynamic simulation technology to obtain an optimum propulsion speed of $0.3 \, \text{m} \cdot \text{s}^{-1}$. In the output process, we used finite element numerical simulation technology to verify that the deformation of the cultivation plate and the maximum stress suffered by it could meet the operational requirements. Finally, operation and performance experiments showed that, under the condition of satisfying the allowable amount of positioning error in the horizontal and vertical directions, the horizontal operation speed was $0.2 \, \text{m} \cdot \text{s}^{-1}$, the maximum positioning error was 2.87 mm, the vertical operation speed was $0.3 \, \text{m} \cdot \text{s}^{-1}$, and the maximum positioning error was 1.34 mm. Accordingly, the success rate of the transport system was 92.5–96.0%, and the operational efficiency was 176–317 plates/h. These results proved that the transport system could meet the operational requirements and provide feasible solutions for the automation of plant factory transport equipment.

**Keywords:** plant factory; automatic transport system; structure design; dynamic simulation; positioning accuracy

## 1. Introduction

The decline of arable land per capita is a global concern. Based on the population growth statistics of the Food and Agriculture Organization of the United Nations (FAO), it is projected that agricultural production will have to increase by 50% in 2050 in order to meet the demand on food supply, which poses a great challenge to global food production [1,2]. To seek effective solutions, sustainable alternatives for food production have been sought in terms of cultivation pattern innovation and improved land utilization. Examples include the application of greenhouse cultivation, plant factories, and vertical farms [3–6]. A vertical farm is defined as a multi-layer plant factory [7]. The plant factory cultivation mode, as an agricultural facility that artificially controls the cultivation environment, can control various factors such as indoor lighting [8,9], temperature and humidity [10,11], and carbon dioxide concentration [12], and is capable of providing safe, high-quality, and pollution-free food [13,14]. Multi-layer stereoscopic cultivation is commonly used in plant factories, and its yield per unit area is significantly higher than that of traditional agriculture [15]. However, it is this mode of production that leads to the need for a large

amount of manual involvement in the cultivation processes, especially in the linkage of moving the cultivation plates up and down in the cultivation frames. This operational session is characterized by intensive and repetitive lifting operations and is dangerous for higher cultivation frames [16–18]. Therefore, in order to reduce the production cost of plant factories and also to automate the production systems of plant factories, mechanization of planting transport is an attractive way to improve the efficiency of plant factories and to reduce their costs.

Since the 1990s, researchers from Chiba University in Japan have studied the vertical cultivation mode of plant factories [19]. With the development of large-scale production technology in plant factories, the transport modes of cultivation plates have gone through different technological stages. The earliest and most common transport mode was manual transport using ladders, and later people developed scissor lift carts for transport [20,21]. For example, Yu et al. [22] developed a scissor-type multifunctional operation platform for cultivation beds, which could be used for autonomous trajectory and positioning as well as the moving in and out of cultivation beds. Zhou et al. [23] developed a set of cultivation plate transport equipment with dynamic and static slide rails and a scissor lifting mechanism which could coordinate and position the guide rail and manipulator through programmable logic controllers (PLC), so as to complete the picking up and putting down of cultivation plates. With the continuous promotion of industrial warehousing and logistics technology, researchers have applied warehousing and logistics transport equipment in plant factory transport systems, which greatly improves the efficiency and automation level of transport operations [24,25]. Gu et al. [26] simulated and analyzed four common logistics transport modes, and designed an input and output lift cart with a guide pushing vehicle for use in cultivation plate transport systems, aiming to improve the transport efficiency of cultivation plates and reduce the transport cost. The plant factory lettuce transport system developed by the SANANBIO Company was composed of a shuttle car, a logistics lift cart, and a guiding cart, and could be used for the transport of lettuce fixing plates at any position of the cultivation frame, with a small physical volume and a high degree of automation [27]. In addition, the positioning accuracy, labor efficiency, and resource utilization of a transport system can be improved by refining the positioning method of the transport and plant equipment [28–30] and optimizing the motor control algorithms [31,32], ultimately contributing to the sustainability and productivity of agricultural production systems.

There are some existing challenges for cultivation plate transport systems in plant factories. The production mode of manual cultivation plate transport has high labor intensity, and labor costs are increasing year by year. Although scissor lifts are widely used, they have the disadvantages of posing safety hazards when working at height, poor alignment accuracy, and low transport efficiency. A large-scale plant factory using an automated transport system has the problems of large capital investment, complex equipment operation, and maintenance. Meanwhile, these automated transport devices cannot be directly replicated and used in other plant factories, so more generalized transport equipment is demanded [24].

Therefore, in order to address these problems, including the high labor intensity of plant factory cultivation plate transport operations, the mismatch of transport equipment, and the poor versatility of existing solutions, this study designed a plant factory cultivation plate transport system with the A-type stereoscopic cultivation frame as the research object. In the design of the input actuator, the key factors affecting the stability of the planting cups in the pushing process were derived by force analysis, and the influence characteristics of the pushing speed on the stability of the planting cups in the pushing process were analyzed using dynamic simulation techniques. For the output actuator of the cultivation plate, finite element numerical simulation was used to analyze the deformation and stress distribution of the cultivation plate during the process of picking up the plate, which ensured that the output actuator could meet the design requirements. Then, the operating parameters of the transport system were optimized through a performance test of the transport system,

and the feasibility of the transport system was verified. Finally, the transport system was tested and evaluated via positioning accuracy experiments and transport system success rate experiments, which optimized the operating parameters of the horizontal and vertical motion mechanisms and verified the productivity of the transport system. This study can provide a useful reference for subsequent research on plant factory transport equipment.

## 2. Materials and Methods

### 2.1. Overall Structure and Working Principles of the Cultivation Plate Transport System

2.1.1. Structure of the Stereoscopic Cultivation Frame and Transport Method

The structure of the A-type stereoscopic cultivation frame used in this study is shown in Figure 1. The cultivation frame has three layers and is mainly composed of a frame (1), cultivation troughs (2), cultivation plates (3), planting cups (4), and slides (5). A slide is installed at both ends of each cultivation trough to facilitate the sliding of the cultivation plates in the cultivation troughs. The layer height of the cultivation frame is changeable according to the lighting conditions of the locality. The cultivation frame is designed with a height of 500 mm, a base level of 400 mm from the ground, and a width of 1400 mm. The length of the cultivation plates was 530 mm, and the width was 310 mm. The cultivation plates were designed with handles at both ends for easy gripping.

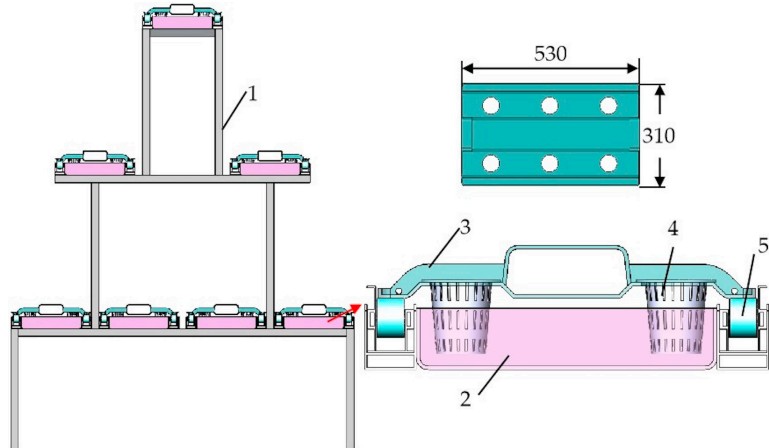

**Figure 1.** Structure of the cultivation frame. 1. Frame; 2. cultivation trough; 3. cultivation plate; 4. planting cup; 5. slide.

Based on the structural analysis of the cultivation frame and the cultivation plates, the cultivation plate transport system in this study can adopt the progressive push transport mode. According to the research of Gu et al. [26], there are two transportation modes suitable for progressive push transport; the first one adopts an input/output lift cart with a pushing guide vehicle to complete the cultivation plate transportation, and the other adopts an input/output lift cart with an interlayer cultivation conveyor to complete the cultivation plate transportation. However, both modes require the installation of motion mechanisms at both ends of the cultivation troughs, which causes high costs and high failure rates. When outputting the cultivation plates, both modes use flat pushing to feed the plates into the lift cart, which results in interference between the fixed-value cups hanging below the plates and the plugs of the cultivation troughs. Therefore, considering the characteristics of the above two transportation modes and the structure of the cultivation frame, this study proposes a conveying method as shown in Figure 2 below. Specifically, a set of input lift carts and a set of output lift carts are installed at both ends of the cultivation frame, and the cultivation plates are pushed by the input actuator in the input lift carts, while the cultivation plates are taken out by the output actuator from the cultivation trough. The input of the next cultivation plate occurs concurrently with the output of the preceding one. This model eliminates the need to install powered devices at both ends of the cultivation

trough and the need to install carts in the cultivation frame, which saves costs and reduces the failure rate of the system.

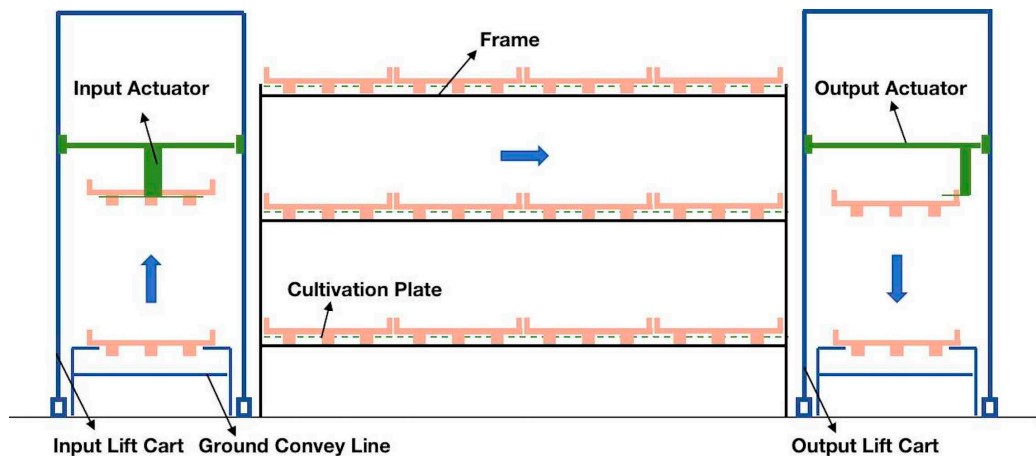

**Figure 2.** Schematic diagram of the cultivation plate transport process.

### 2.1.2. Structure of the Transport System

In accordance with the transporting method described above, this study designed a cultivation plate transport system (see Figure 3) which mainly consists of conveying lines (1), lift carts (2), input actuators (3), cultivation frames (4), output actuators (5), and ground guide rails (6). The cultivation plate transporting system can be divided into the input system and the output system, according to their usage. The input system consists of a set of lift carts and input actuators, and the output system consists of a set of lift carts and output actuators. Both systems are arranged at the ends of the cultivation frame with a ground conveyor line arranged directly below each of them.

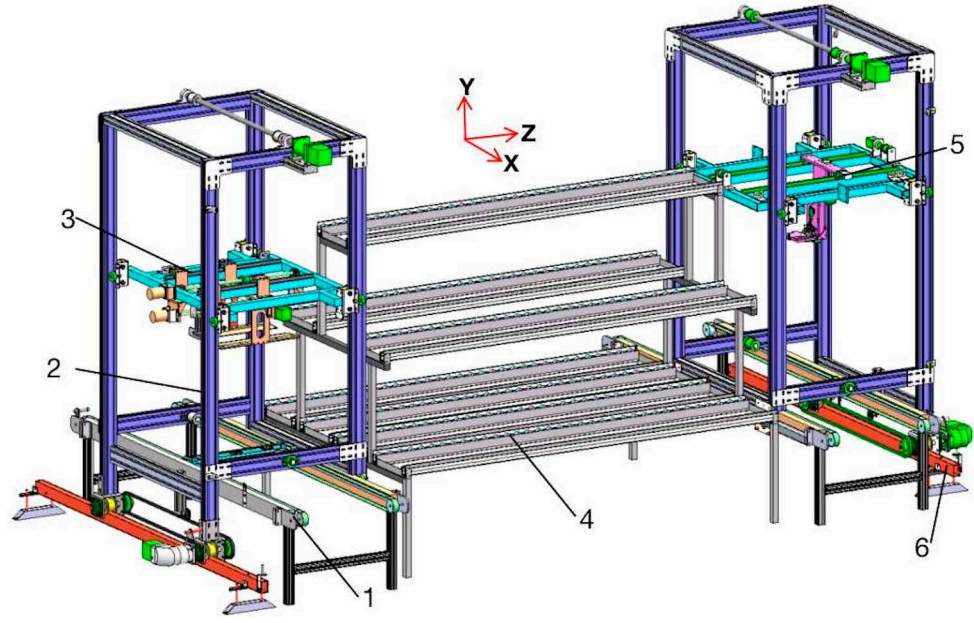

**Figure 3.** Structure of the transport system. 1. Ground conveyor line; 2. lift cart; 3. input actuator; 4. cultivation frame; 5. output actuator; 6. ground guide rail.

### 2.1.3. Working Principle

The working principle of the input and output of the cultivation plate are as follows:

The transport system allows movement in the horizontal (X-axis) and vertical (Y-axis) directions, which is accomplished by a lift cart. The input and output actuators are responsible

for operating in the Z-axis direction. Each cultivation frame is equipped with a position sensor that helps the lift cart to recognize the target position. For the first input, the lift cart moves to the target cultivation frame and stops. Then, the cultivation plate on the ground conveyor line triggers the sensor on the lift cart, at which point the cultivation plate coincides with the position of the target cultivation trough. The input actuator then grabs it and places it in the cultivation trough. The system repeats this process until one cultivation trough is full. When removing the cultivation plate, the input actuator and the output actuator have to work together. First, the output actuator grips the cultivation plate handle and, with the help of the lift cart, removes the cultivation plate and places it on the ground conveyor line. At the same time, the input actuator grabs a new cultivation plate and puts it into the cultivation trough, so that the output actuator can continue to pick up the next one. The system repeats this operation to achieve continuous transport operation.

### 2.2. Structural Design of the Lift Cart

The motion mechanism of the lift cart contains two sections, the horizontal motion mechanism and the vertical motion mechanism, where their main function is to realize the precise conveyance of the cultivation plates. The structure of the lift cart is shown in Figure 4. The horizontal motion mechanism and the vertical motion mechanism are installed on the frame. As the carrier of all components, the frame plays a vital role in the overall stability. Therefore, the frame in this study is made of industrial aluminum profile, which is light in quality and good in rigidity. The cross-section size of the aluminum profile is $40 \times 80$ mm and the external dimensions of the lift cart are $910 \times 1100 \times 2130$ mm. To ensure the accuracy of the transport system, both the horizontal and vertical drive motors use servo motors, and the Y-axis drive motors are equipped with a self-locking function. After careful estimation, the total mass of the whole machine ($M_1$) is no more than 100 kg, and the vertical direction synchronous belt load ($M_2$) is no more than 35 kg, so chain transmission was selected for horizontal motion and synchronous belt transmission was selected for vertical motion. In accordance with the above load information and the requirements of transmission parts, the main component design parameters of the horizontal and vertical motion mechanism are shown in Table 1.

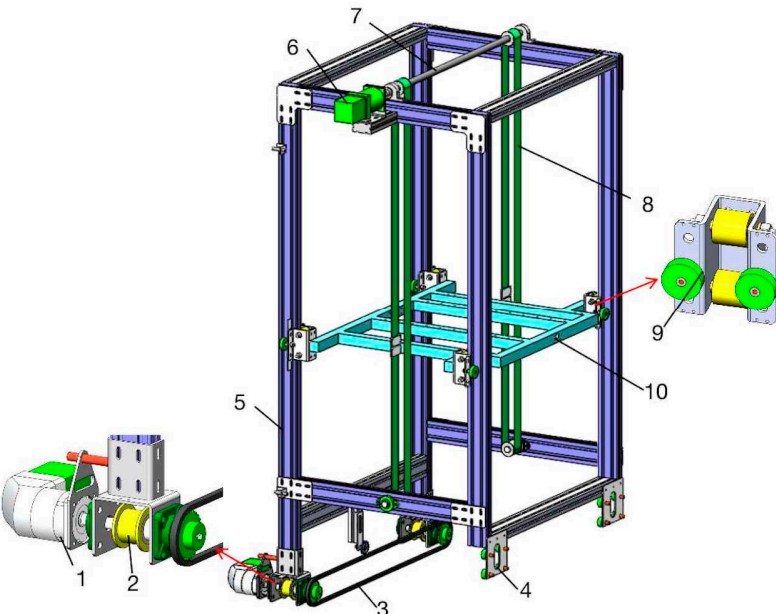

**Figure 4.** Structure of the lift cart. 1. Horizontal drive motor; 2. riving wheels; 3. chain transmission components; 4. driven wheel; 5. Frame; 6. vertical drive motor; 7. driving shaft; 8. synchronous belt drive components; 9. lifting guide wheel; 10. lift platform.

**Table 1.** Parameters of the main components of lift carts.

| Main Components | Parameters |
|---|---|
| Horizontal drive motor | Power $P$ = 0.75 kW, rated torque $T$ = 2.39 N.m, rated speed $n$ = 3000 rpm. |
| Vertical drive motor | Power $P$ = 0.75 kW, rated torque $T$ = 2.39 N.m, rated speed $n$ = 3000 rpm. |
| Driving wheels | Diameter $D_1$ = 60 mm, width = 42 mm. |
| Chain transmission component | Sprocket pitch $P$ = 9.525 mm, distribution circle diameter = 82 mm, 27 teeth. Chain type = 06B. |
| Synchronous belt drive components | Belt wheel: pitch $P$ = 5 mm, 33 teeth, pitch circle diameter $D_2$ = 52 mm. Synchronization bandwidth = 50 mm, length = 3410 mm. |
| Driving shaft | Diameter = 25 mm, the length = 1070 mm. |
| Inductive proximity switches | Thread diameter = M12, PNP normally open, working voltage = DC 10–30 V, maximum detection distance = 15 mm, model = I2B1204NO. |

According to the design parameters in Table 1, the design power of the drive motors in the horizontal and vertical directions was estimated. The motor power calculation formula is as follows.

$$P_d = \frac{P_t \cdot k}{\eta} \tag{1}$$

where $P_d$ represents the actual required power of the motor, in W; $P_t$ is the ideal power, in W; $k$ is the overload coefficient, and 2; $\eta$ is the motor efficiency, 0.9 [33].

The ideal power of the motor is calculated as follows:

$$P_t = \frac{T}{i} \times \frac{n}{9550} \tag{2}$$

where $T$ is the load torque, in N.m; $i$ is the motor reduction ratio, 10; and $n$ is the motor speed, 3000 rpm.

According to the horizontal and vertical direction motion analysis, the load torques in the two directions are:

$$\begin{cases} T_1 = \frac{D_1}{2} \times \mu M_1 g \\ T_2 = \frac{D_2}{2} \times M_2 g \end{cases} \tag{3}$$

where $T_1$ is the horizontal load torque, in N.m; $T_2$ is the vertical load torque, in N.m; $M_1$ is the total weight of the single-side whole machine, 100 kg; $M_2$ is the maximum weight in the vertical direction, 35 kg; $D_1$ is the diameter of the driving wheel, 60 mm; $D_2$ is the diameter of the belt pulley pitch circle; 52 mm, $\mu$ is the rolling friction coefficient, 0.05; and $g$ is gravitational acceleration, 9.8 m·s$^{-2}$.

Calculated from Equations (1)–(3), the actual power required for the horizontal drive motor is 0.10 kW, and the actual power required for the vertical upgrading drive motor is 0.62 kW. Therefore, the selections for the horizontal drive motor and the vertical drive motor meet the design requirements.

*2.3. Design of Key Components of the Input Actuator*

2.3.1. Structural Design

The structure of the input actuator is shown in Figure 5 below, and its main function is to push the cultivation plate into the cultivation trough after grasping it. Based on its functions, the input actuator can be divided into the primary propulsion mechanism, the secondary propulsion mechanism, and the clamping mechanism. The primary propulsion mechanism contains a primary propulsion motor (1), a one-way propulsion screw pair (2),

sliding rails and sliders (3), and a U-shaped connecting plate (4), which are fixed above the lift platform. The secondary propulsion mechanism contains a secondary propulsion motor (5), a secondary propulsion screw pair (6), and a knockout plate (7), which are suspended below the U-shaped connecting plate (4) and are parallel to the installation direction of the primary propulsion mechanism. The clamping mechanism contains a clamping motor (8), a clamping screw pair (9), and a clamping plate (10), which are suspended below the U-shaped connecting plate (4) and are perpendicular to the direction of the secondary propulsion mechanism.

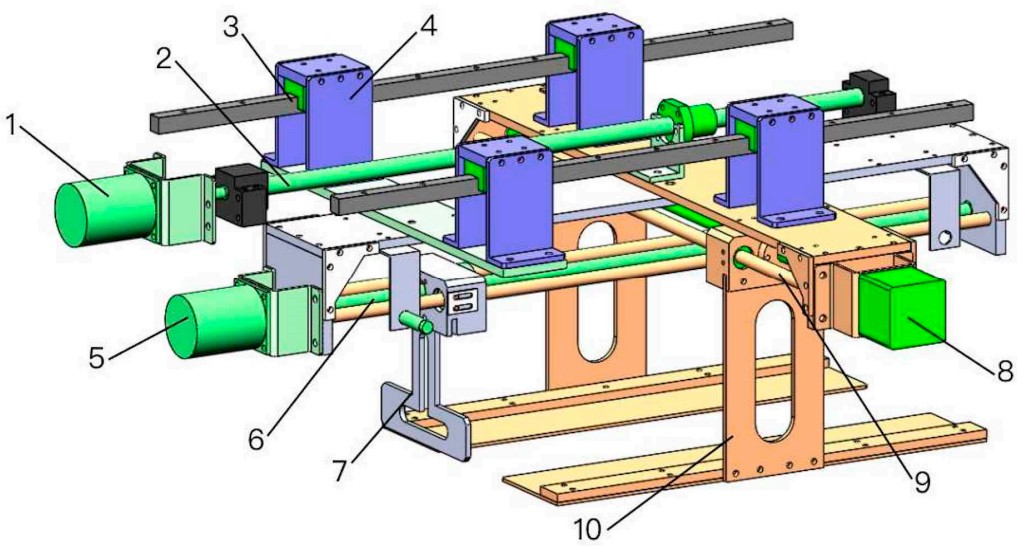

**Figure 5.** Structure of the input actuator. 1. Primary propulsion motor; 2. primary propulsion screw pair; 3. slide rails and sliders; 4. U-shaped connecting plate; 5. secondary propulsion motor; 6. secondary propulsion screw pair; 7. knockout plate; 8. clamping motor; 9. clamping screw pair; 10. clamping plate.

The structural parameters of the ball screw are essential for the motion and positioning of the input actuator. If the diameter of the ball screw is too large, it will increase the overall size of the entire motion mechanism, resulting in material waste and increased costs, while if the diameter is too small, it will lead to problems such as jittering and damage to the screw, which will affect the clamping and propulsion operations. After reviewing the parameter information, the structural parameters of the ball screw used for the propulsion motion were initially set as a diameter of 16 mm and a lead of 10 mm. The propulsion motor that was selected is a DC speed control motor with a rated speed of 3000 rpm and a rated torque of 0.32 N.m. The clamping mechanism utilizes a bi-directional ball screw with a diameter of 12 mm and a lead of 10 mm. In the case of the clamping motor, a stepping motor was selected, with a torque of 1.2 N.m and a clamping speed of 50 mm s$^{-1}$. To ensure the reliability of the selected ball screw in the propulsion operation, it is necessary to validate the allowable limit load and allowable limit speed of the ball screw. The axial allowable limit load and limit speed of the ball screw is calculated as follows.

$$\begin{cases} P_c = \rho \dfrac{d_r^4}{k^2} \times 10^4 \\ N_c = \gamma \dfrac{d_r}{k^2} \times 10^7 \end{cases} \tag{4}$$

where $P_c$ is the allowable limit load, in N; $N_c$ is the allowable limit speed of the ball screw, in rpm; $k$ is the spacing between the nut and the support seat, 556 mm; and $d_r$ is the inner diameter of the ball screw, set as 13.5 mm. $\rho$ and $\gamma$ are the coefficients determined by the support method of the ball screw; $\rho$ is taken as 10, and $\gamma$ is 15.4 [34].

Substituting the relevant parameters of the propulsive motion into the above equation, it can be concluded that $P_c = 10{,}744.5$ N, and $N_c = 6725.2$ rpm. The actual rotational speed

of the propulsive motor in this study is less than 3000 rpm, so the selected screw meets the design requirements. The load of the propulsion movement mainly contains two parts: its weight and the resistance force of the disk-pushing process. According to calculations made using SolidWorks software (version 2020), the maximum weight of the input actuator is not more than 20 kg. Since each cultivation trough in this study can hold up to 38 cultivation plates, we measured the maximum push resistance of these plates to be 26.64 N and the friction generated by their weight to be 3.92 N (with a combined friction coefficient of 0.02). Therefore, the maximum axial load is $F$ = 30.56 N. The torque required to overcome the load is:

$$T = \frac{F \cdot P}{2\pi\xi} \tag{5}$$

where $T$ is the axial load torque, in N.m; $F$ is the maximum axial load, 30.56 N; $P$ is the screw lead, 10 mm; and $\xi$ is the ball screw efficiency, 0.9.

After calculation, the torque $T$ = 0.05 N.m is required for the propulsion to overcome the axial load; therefore, the torque of the selected propulsion motor meets the design requirements.

### 2.3.2. Parameter Setting of Clamping and Pushing Mechanism

The cultivation plate clamping process requires that the cultivation plate and the planting cups be lifted at the same time, and that the length of the clamping plate should be smaller than the length of the cultivation plate. The parameter settings of the clamping mechanism are shown in Figure 6 below.

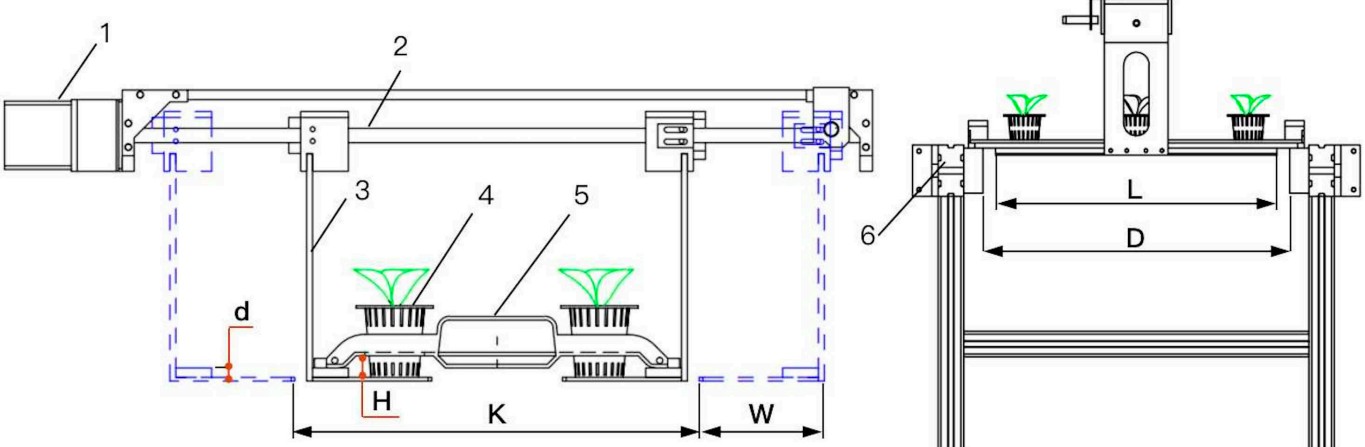

**Figure 6.** Diagram of clamping mechanism operation. 1. Clamping motor; 2. clamping screw pair; 3. clamping plate; 4. planting cup; 5. planting plate; 6. ground conveyor line.

To enable the cultivation plate and planting cups to be lifted at the same time, the width of the bottom of the clamping plate $W$ is set to 100 mm and the opening width of the clamping mechanism $K$ is set to 330 mm, so the clamping displacement of the screw is 110 mm. The length of the clamping plate $L$ should meet the requirement of lifting three planting cups at the same time and should be less than the width of the ground conveyor line between the belts $D$, so the length of the clamping plate $L$ is set to 440 mm. After calculation, when the lifted cultivation plate is flush with the upper surface of the cultivation plate in the cultivation frame, the height of the lifted cultivation plate is 8 mm, the planting cup support height $H$ is 13 mm, and the total length of the clamping screw is 640 mm.

As shown in Figure 7, the processes of introducing the cultivation plate into the cultivation trough can be divided into two stages: primary propulsion and secondary propulsion. First, the cultivation plate is raised to a predetermined height of the target cultivation trough under the control of the clamping mechanism (Figure 7a). Then, the

primary propulsion mechanism pushes the cultivation plate to the right, the cultivation plate on the clamping plate collides with the cultivation plate in the cultivation trough, and then moves to the right together until the clamping plate is close to the end before stopping, which is the primary propulsion process (Figure 7b), with a displacement $S_1$ of 230 mm. Finally, the secondary propulsion mechanism pushes the cultivation plate inside the clamping mechanism to the target cultivation trough, which completes the secondary propulsion process (Figure 7c), and its displacement $S_2$ is 483 mm.

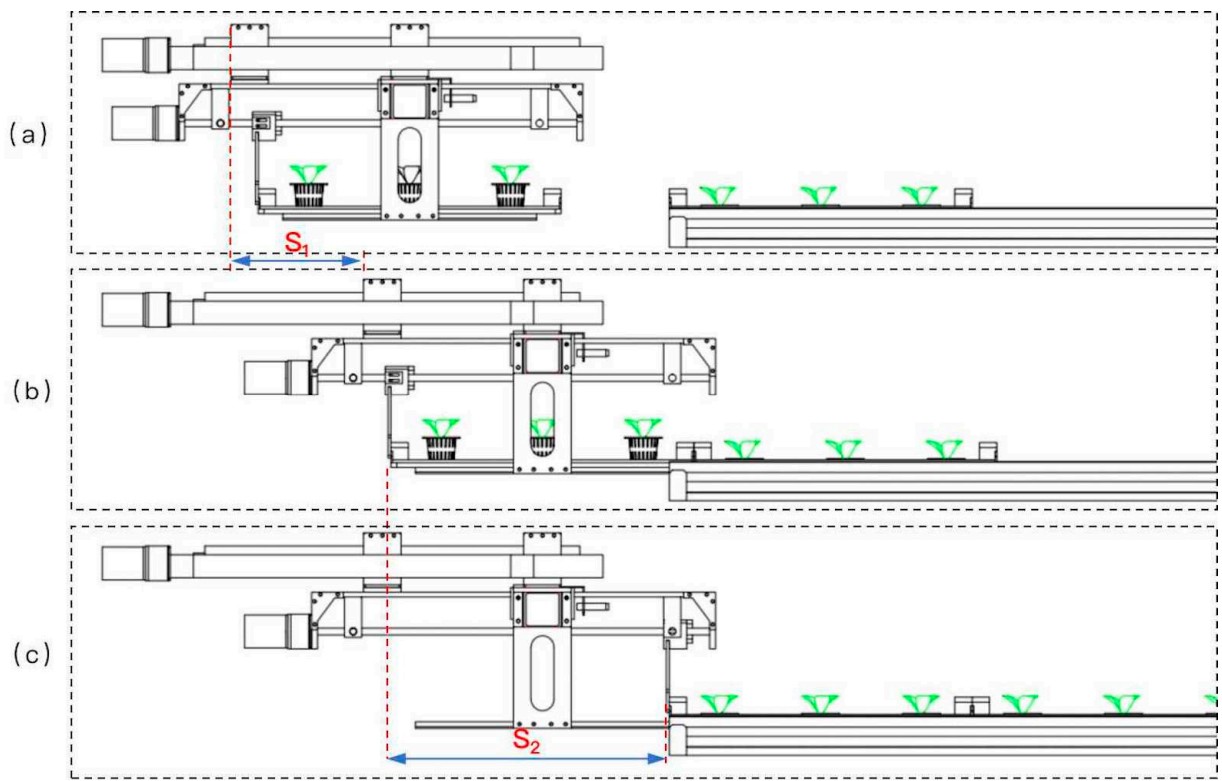

**Figure 7.** Schematic diagram of processes of pushing the cultivation plate into the cultivation trough; (**a**) shows the initial position before pushing the plate; (**b**) shows the primary propulsion process; (**c**) shows the secondary propulsion process.

### 2.3.3. Force Analysis of the Collision Processes of the Planting Cups

As shown in Figure 8, a gap between a planting cup and hole of the cultivation plate is created because the planting cup is lifted to a certain height. During the primary propulsion process of the cultivation plate, the cultivation plate and the planting cups collide with the cultivation plate in the cultivation trough at a certain speed. This collision process may cause the planting cups to tip over. The following force analysis of a planting cup was performed.

The threshold condition for the overturning of the planting cup is that the bottom pivot point $N$ coincides with the line of gravity of the planting cup. Assuming that the planting cup is in a critical state of imminent overturning, it is possible to establish the vertical and horizontal coordinate axes, which can be obtained by D'Alembert's principle:

$$\begin{cases} F_I + F_{N2} \cos\alpha = f_2 \sin\alpha + f_1 \\ F_{N2} \sin\alpha + f_2 \cos\alpha = G - F_{N1} \\ (G - F_{N1}) \cdot H \tan\alpha + F_I(L - H) + f_1 \cdot H \end{cases} \tag{6}$$

To simplify the calculation of the system of equations, assuming that the friction coefficients of both $f_1$ and $f_2$ are $\mu$, then $f_1 = \mu F_{N1}$ and $f_2 = \mu F_{N2}$, which can be obtained by bringing them into Equation (6):

$$\alpha = arc \tan \sqrt{\frac{F_I[(L-H)(1+\mu^2)+\mu^2 H]+\mu G H}{H(F_I - \mu G)}} \qquad (7)$$

According to Equation (7), it can be seen that the overturning angle of the planting cup is governed by a variety of factors, including the inertia force $F_I$, the height of the center of gravity $L$, the height of the support of the planting cup (13 mm), the gravitational force of the planting cup $G$, the coefficient of friction $\mu$, and so on. $L$, $G$, and $\mu$ can be measured and queried to obtain specific values. Therefore, the main factor determining the overturning of the planting cup is the magnitude of the inertia force $F_I$. The propulsion speed of the propulsion mechanism is related to the magnitude of $F_I$, and the propulsion speed can be appropriately reduced to ensure that the planting cups will not tip over during the propulsion process.

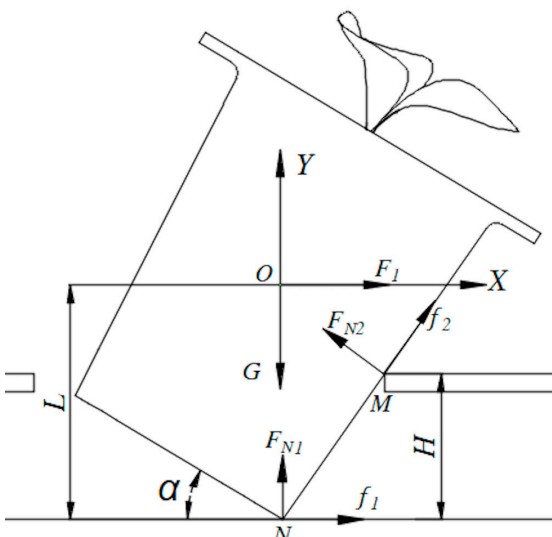

**Figure 8.** Force analysis of a planting cup. $F_I$ is the inertia force, in N; $F_{N1}$ is the support force of the clamping plate on the planting cup, in N; $F_{N2}$ is the support force of the cultivation plate on the planting cup, in N; $f_1$ is the friction force of the clamping plate on the planting cup, in N; $f_2$ is the friction force of the cultivation plate on the planting cup, in N; $G$ is the gravitational force of the planting cup, in N; $H$ is the planting cup support height, in mm; $L$ is the height of the center of gravity of the planting cup, in mm; $\alpha$ is the turning angle of the planting cup, in rad.

2.3.4. The Control Flow and the PLC Circuit of Cultivation Plate Input and Output Actuators

According to the above working principle, this study designed a control flow chart of the input and output actuators, and built a hardware control cabinet for the control system. The input control system includes servo motors for horizontal and vertical movement, clamping motors, propulsion motors, and proximity switches. The output control system includes servo motors for horizontal and vertical movement, propulsion motors for the grabbing mechanism, screw motors, electric push rods, and proximity switches. Due to the length of this section, detailed control flow charts, key parts of circuit diagrams, and parameters of all electrical components are given in the Supplementary Material.

*2.4. Design of Key Components of the Output Actuator*
2.4.1. Structural Design and Working Principle

As shown in Figure 9, the actuator mainly consists of a stepping motor (1), a support plate (2), a lift platform (3), synchronous belt drive components (4), a sliding rails and sliders for propulsion (5), and a grabbing mechanism (6). The grabbing mechanism, as the core of the output actuator, contains a rotating gripper (7), a clamping push plate (8), a rack and pinion (9), rotating slide rails and sliders for rotating sliders (10), a connecting rod (11),

an electric pushing rod (12), a screw stepper motor (13), a telescopic push plate (14), a vertical fixing plate (15), a telescopic slide rails and sliders for rotating sliders (16), and a horizontal support plate (17). The stepping motor, synchronous belt drive components, and support plate sliding rails and sliders for propulsion together form a propulsion mechanism for the input actuator and are mounted above the lift platform. The grabbing mechanism is connected to the propulsion mechanism through the support plate and is capable of extending and retracting with the support plate on the propulsion slide. The function of the grabbing mechanism can be divided into two parts: one is the linear reciprocating motion part, which adopts the silk rod slider mechanism; and the other is the rotary motion part, which adopts the connecting rod slider mechanism.

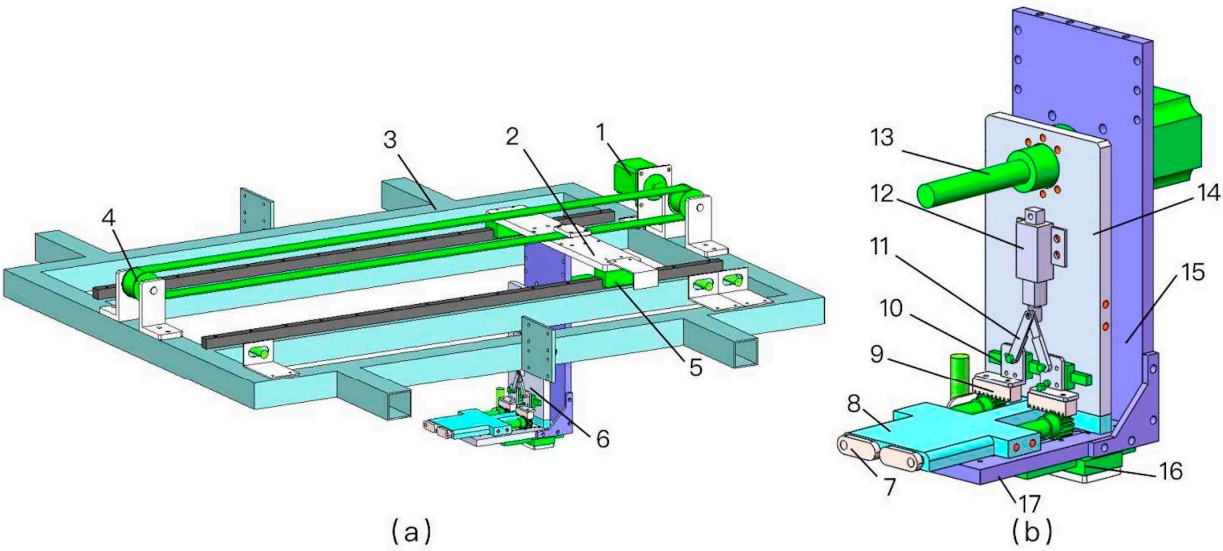

**Figure 9.** Schematic diagram of output actuator structure. (**a**) Structure of the output actuator; (**b**) the grabbing mechanism.

The cultivation plate output workflow is as follows: first, the propulsion mechanism is activated, and the synchronous belt drives the grabbing mechanism to move in the direction of the target cultivation trough and stops when the clamping push plate reaches the handle of the cultivation plate (Figure 10a). Next, the screw stepper motor of the grabbing mechanism rotates, and the clamping push plate continues to reach into the handle of the cultivation plate for a certain distance to ensure that the rotating gripper can completely pass through the handle (Figure 10b). Then, the electric pushing rod extends and moves the rack and pinion through the connecting rod to rotate the gripping gripper 90 degrees (Figure 10c). Next, the screw stepper motor reverses to retract the clamping push plate, while the clamping gripper pulls the cultivation plate back and secures it to the front of the horizontal support plate, completing the clamping operation of the cultivation plate (Figure 10d). Finally, the Y-axis drive motor lifts the cultivation plate out of the cultivation trough, the Y-axis drive motor of the propulsion mechanism reverses to retract the whole body to stop right above the ground conveyor line, and then the whole body descends to place the cultivation plate inside the conveyor line. After placing, the rotating gripper rotates 90 degrees, and the plate-grabbing device exits the cultivation plate handle, completing a plate-taking operation.

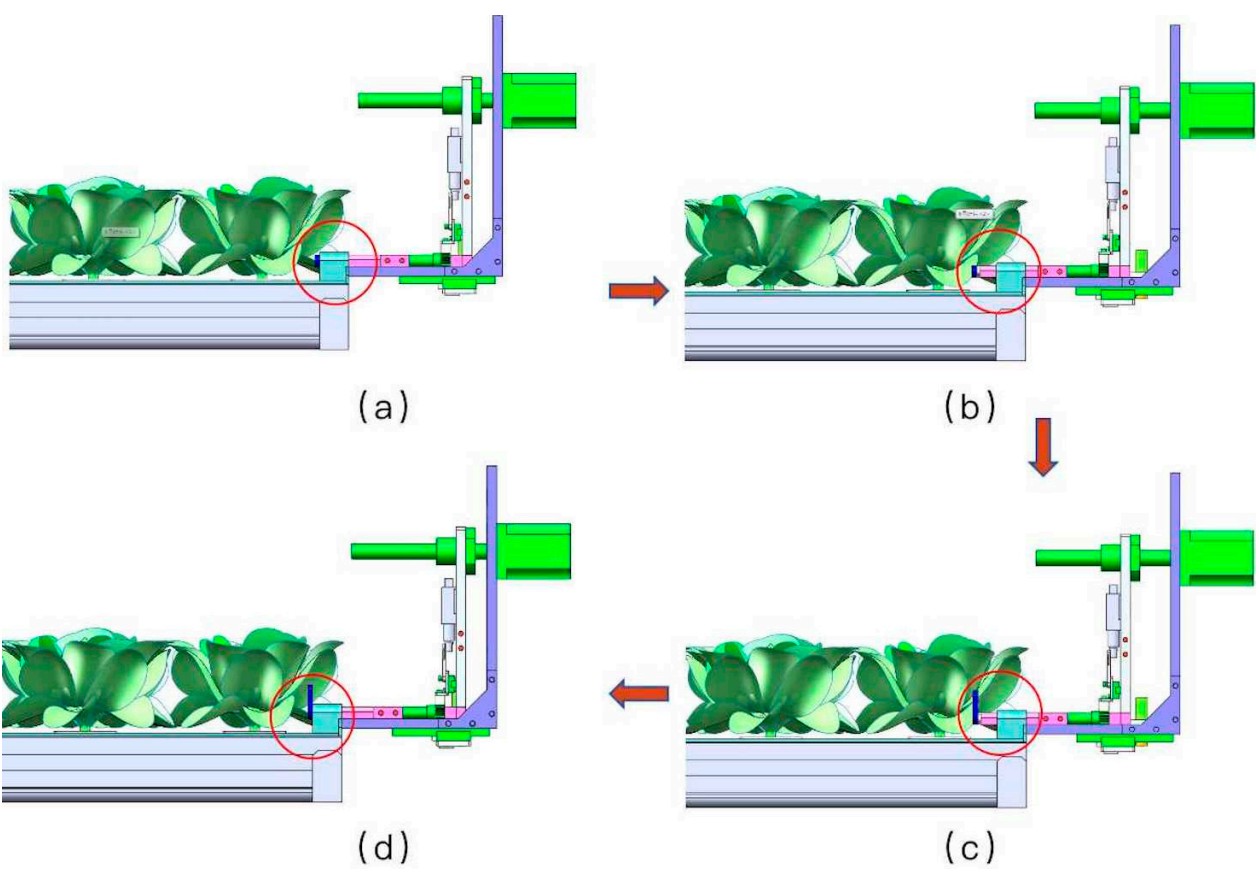

**Figure 10.** Process of gripping cultivation plate. Note. The red circles show the movement change area of the rotary gripper.

### 2.4.2. Parameter Design of Linear Reciprocating Motion Components

The main function of the linear reciprocating motion is to reach inside the handle of the cultivation plate and fix it to the front of the horizontal support plate. The parameter design of the related parts is shown in Figure 11. According to measurements, the maximum width inside the handle of the cultivation plate is 95 mm and the internal height is 37 mm. To meet the clamping function requirements, the width of the clamping push plate is designed to be 80 mm, with a thickness of 12 mm, and a reserved error tolerance of 7.5 mm at the left and right ends. The width of the horizontal support plate is designed to be 120 mm (greater than the maximum width of the handle), with a thickness of 12 mm. In order to increase the clamping stability, two sets of rotating grippers are designed with a length of 35 mm and a width of 10 mm. In the process of putting the cultivation plate into the conveyor line, in order to avoid interference with the skeleton of the conveyor line, the distance of the upper surface of the clamping push plate from the top of the handle is set to be 5 mm.

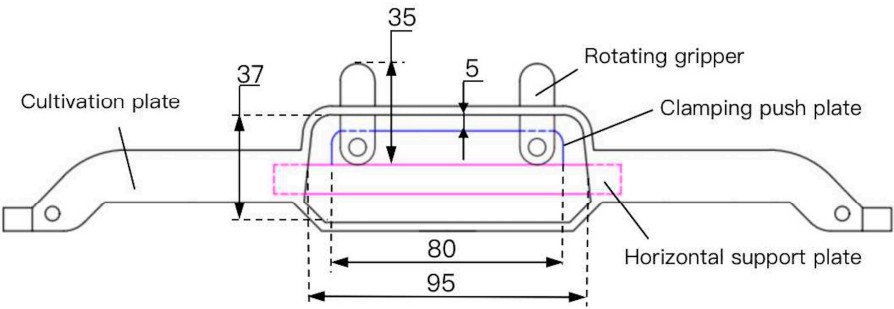

**Figure 11.** Parameter settings for linear reciprocating motion.

### 2.4.3. Parameter Design of Rotating Parts

The function of the rotating component is to rotate the rotating gripper by 90 degrees to achieve stable fixation of the cultivation plate. The installation position of each part is shown in Figure 12a. The rotating grippers are connected to the gears, and the electric pushing rod drives the sliders and racks to carry out linear reciprocating motion, thus realizing the rotation of the rotating grippers. Given the requirements of installation space and motion accuracy, the gear indexing circle diameter selected in this study is 18 mm and the modulus is 1. Therefore, when the toggle rod is rotated by 90 degrees, the running displacement of the rack and pinion is $c_1 = 14.1$ mm. The motion process of the connecting rod is shown in Figure 12b, the solid line represents the initial position of the connecting rod, and the blue dashed line indicates the position of the connecting rod after the rotating gripper is rotated by 90 degrees. In this study, the electric pushing rod was preset to have a stroke of $c_3 = 10$ mm and a pushing force of 40 N. According to the geometrical relationship in Figure 12a, $c_2$ is the largest when the connecting rod is horizontal; therefore, at the maximum size of the connecting rod and the slider, it cannot be more than half of the width of the telescopic push plate (106 mm). Therefore, this study set $c_2 = 35$ mm. CAD software (version 2020) calculations yielded $c_4 = 12.6$ mm, $c_5 = 22.7$ mm, and connecting rod clamp angle $\beta$ ranging from 21.1° to 49.7°. These data were imported into SolidWorks for simulation and design, and the installation position of the electric pushing rod could be determined.

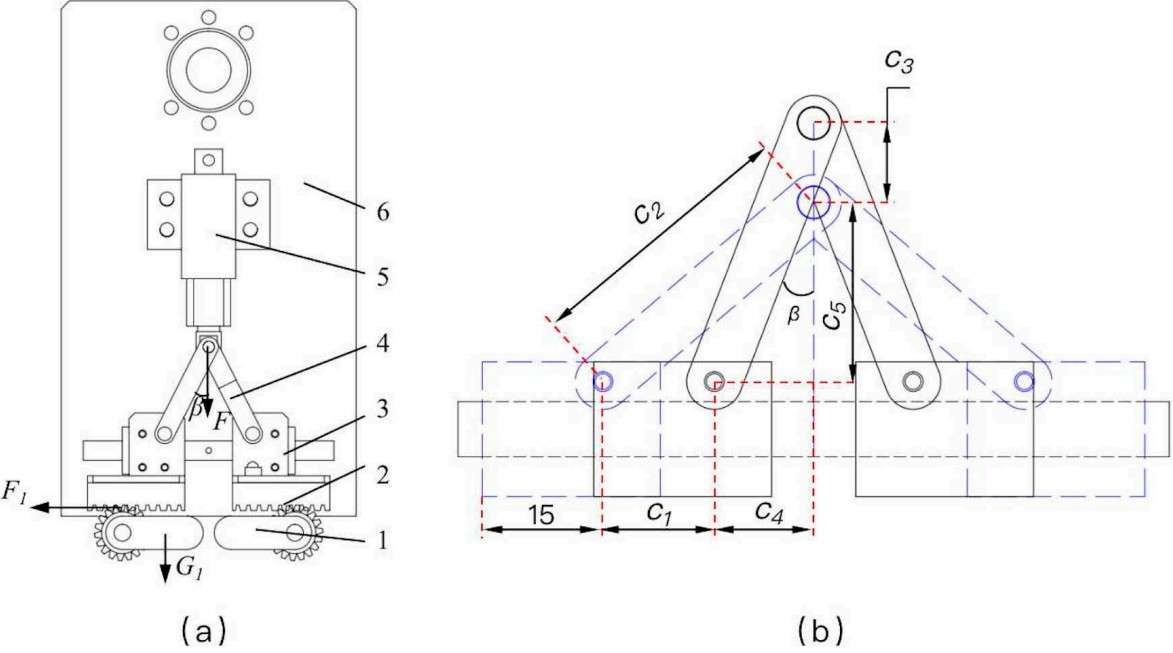

**Figure 12.** Structure of rotating parts. (**a**) Front view of rotating parts; (**b**) connecting rod motion process. 1. Rotating gripper; 2. rack and pinion; 3. rotating slide rails and sliders; 4. connecting rod; 5. electric pushing rod; 6. telescopic push plate.

Once the connecting rod clamp angle is determined, the actual thrust required by the actuator can be estimated. The mass of the rotating gripper is about 14.1 g and the distance from the center of gravity to the rotating axis ($l_1$) is about 11.5 mm. From the force analysis of the rotating mechanism in Figure 12a, it can be seen that the thrust force ($F$) required by the electric pushing rod satisfies the following equation.

$$\begin{cases} G_1 l_1 = \dfrac{F_1 d}{2} \\ F = \dfrac{2F_1}{\eta_1 \eta_2 \tan \beta} \end{cases} \tag{8}$$

where $G_1$ is the magnitude of gravity of rotating gripper, in N; $F_1$ is the horizontal tangential force acting on the gear, in N; $d$ is the diameter of the gear indexing circle, 18 mm; $\eta_1$ is the efficiency of sliding rails and sliders, 0.99; $\eta_2$ is the rack and pinion meshing efficiency, 0.95; $F$ is the thrust force required by the electric pushing rod, in N; and $\beta$ is the connecting rod clamp angle, in °.

By substituting the relevant data into Equation (8), the maximum thrust required by the electric pushing rod is calculated as $F = 0.97$ N. The thrust of the actual selection for the electric pushing rod can reach 40 N, which is sufficient for the actual working requirements.

2.4.4. Force Analysis of the Cultivation Plate during Gripping Process

During the gripping process, the cultivation plate is subjected to unilateral force and the stress is mainly concentrated in the handle area. To investigate whether this process leads to damage to the cultivation plate, this study carried out a force analysis of the cultivation plate during the gripping process, as shown in Figure 13.

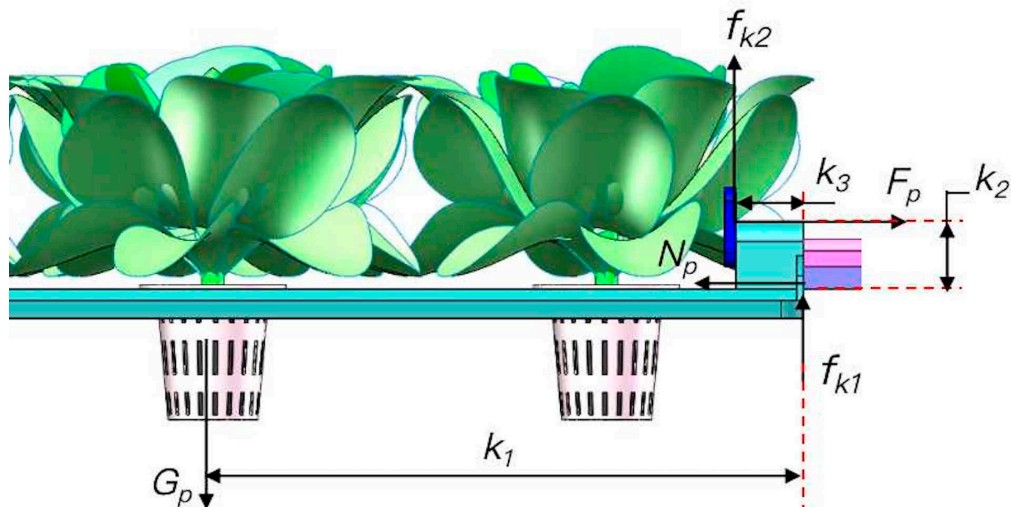

**Figure 13.** Force analysis of cultivation plate.

The force balance equation for a cultivation plate in a balanced state is as follows:

$$\begin{cases} F_p = N_p \\ G_p = f_{k1} + f_{k2} \\ G_p k_1 = F_p k_2 + f_{k2} k_3 \\ f_{k1} = f_{k2} \end{cases} \quad (9)$$

where $G_p$ is the gravitational force of the cultivation plate with mature lettuce, in N; $k_1$ is the distance from the gravitational force $G_p$ to the support point, in mm; $F_p$ is the pulling force of the rotating gripper on the cultivation plate, in N; $k_2$ is the distance from the pulling force F to the support point, in mm; $k_3$ is the distance from the static friction force $f_{k2}$ to the support point, in mm; $N_p$ is the support force of the horizontal support plate on the cultivation plate, in N; and $f_{k1}$ and $f_{k2}$ are the static friction forces, in N.

Measurements of the model in SolidWorks yielded $k_1 = 262.5$ mm and $k_2 = k_3 = 30$ mm. Actual measurements showed that the weight of a single ripe lettuce and its planting cup was about 285 g, the weight of the cultivation plate was 685 g, and the total weight of the cultivation plate and the planting cups containing the ripe lettuce was about 2.34 kg. The gravitational acceleration g was taken to be 9.8 m·s$^{-2}$. Substituting this data into Equation (9), it was calculated that $N_p = F_p = 88.86$ N. The required pulling force when the cultivation plate is gripped is provided by the screw stepper motor. The screw stepper motor selected in this study has a screw diameter of 16 mm and a lead of 5 mm. The model

of the screw stepper motor is 57BYGB56P280, and the holding torque of the screw stepper motor is 1.3 N.m. The total axial load of the screw stepper plate is:

$$F_t = F_p + m_3 g \mu_3 \tag{10}$$

where $F_t$ is the total axial load of the wire rod, in N; $F_p$ is the tension force on the cultivation plate, in N; $m_3$ is the total mass of the wire rod load, 5.14 kg; $\mu_3$ is the coefficient of friction, 0.05, and $g$ is the acceleration of gravity, 9.8 m·s$^{-2}$.

By substituting Equation (10) into Equation (5), it can be calculated that the torque required for the screw stepper motor to overcome the load is 0.08 N.m. The torque of the selected motor is three times greater than the torque required to overcome the load, so the selected screw stepper motor meets the working requirements.

### 2.5. Performance Experiments of the Transport System

For the operation performance of the transport system, this study conducted experiments on the positioning accuracy of its horizontal running mechanism (X-axis) and vertical motion mechanism (Y-axis). The positioning errors at different speeds were measured and then analyzed to obtain the optimal operating speed. Under the condition of optimal operating speed, the success rate and efficiency of transporting each layer were tested to verify the feasibility of the design of the transport system.

#### 2.5.1. Positioning Accuracy Experiment Site and Equipment

The experiments were conducted at the Beijing Academy of Agricultural and Forestry Sciences, located at the coordinates 39.95 °N latitude and 116.28 °E longitude. The experiments are shown in Figure 14 below, and the experimental equipment consisted of a lift cart, a ground conveyor line, a tape measure (0–5 m with an accuracy of 1 mm.), a stopwatch, and a laser pointer.

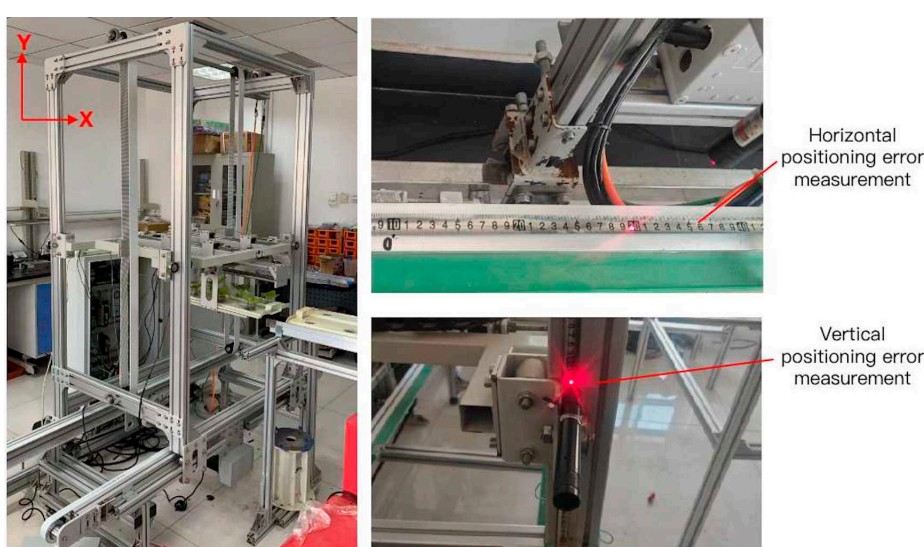

**Figure 14.** Positioning accuracy experiment site.

#### 2.5.2. Experiment Method for Positioning Error

The sequence of operation of the transport system is shown in Figure 15 below, and the transport operation was carried out in the order of the numbers (1)–(7). There were three kinds of theoretical displacements of the transport system in the horizontal direction, and the target values were 180 mm, 359 mm, and 718, mm respectively. The theoretical lifting height in the vertical direction was the distance from the ground conveyor line to the bottom of each layer of the cultivation plate, and the theoretical lifting heights of the bottom, middle, and top layers were 114 mm, 799 mm, and 1484 mm in that order. According to the

previous experiments, the maximum speed of horizontal and vertical motion mechanism should not exceed 0.4 m·s$^{-1}$; therefore, the running speeds were 0.1, 0.2, 0.3, and 0.4 m·s$^{-1}$. The equipment started from a fixed origin, stopped after reaching the set-theoretical displacement, recorded measured value of the laser pointer's position, corresponding to the position of the equipment when it stopped, and repeated the experiment three times for each group. The average value of these three repeats was taken as the actual measured displacement of the group. The calculation formulas for the relative errors in the horizontal and vertical directions are as follows.

$$\begin{cases} R_x = \frac{|\bar{x}-x|}{x} \times 100\% \\ R_y = \frac{|\bar{y}-y|}{y} \times 100\% \end{cases} \tag{11}$$

where $R_x$ is the relative error in the horizontal direction, in %; $R_y$ is the relative error in the vertical direction, in %; $x$ is the theoretical displacement in the horizontal direction, in mm; $\bar{x}$ is the horizontal positioning error, in mm; $y$ is the theoretical displacement in the vertical direction, in mm; and $\bar{y}$ is the vertical positioning error, in mm.

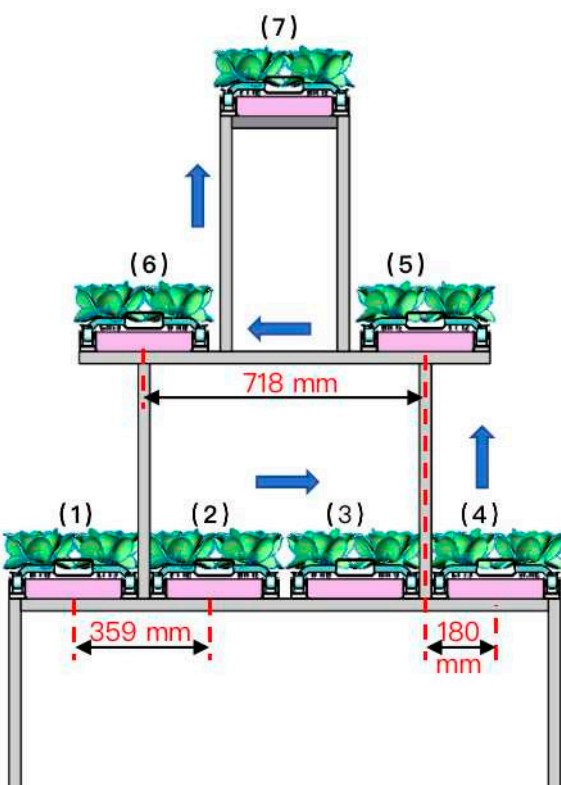

**Figure 15.** Schematic diagram of the sequence of transport operations.

### 2.5.3. The Success Rate and Efficiency of Transport System

After the positioning accuracy test, this study tested the combined operational success and efficiency of the transport system. A group of cultivation frames was selected as the experiment group, and it was ensured that there were enough cultivation plates on the supply side of the ground conveyor line. Each group of cultivation troughs was subjected to 50 input and output operations, and successful completion of both input and output was counted as 1 success; if either part failed, the experiment failed. Even if the transport step fails, the system still needs to complete all the steps to calculate the efficiency. The experimental equipment consisted of a cultivation plate transport system, ground conveyor line, cultivation frame, and stopwatch. The experiment site is shown in Figure 16. The

length of time to completion and number of successes for each layer during the experiment were recorded, and the formula for calculating the success rate of each layer was as follows:

$$R = \frac{N_s}{N} \times 100\% \tag{12}$$

where $R$ is the success rate, in %; $N$ is the total number of experiments, and $N_s$ is the number of successful trials.

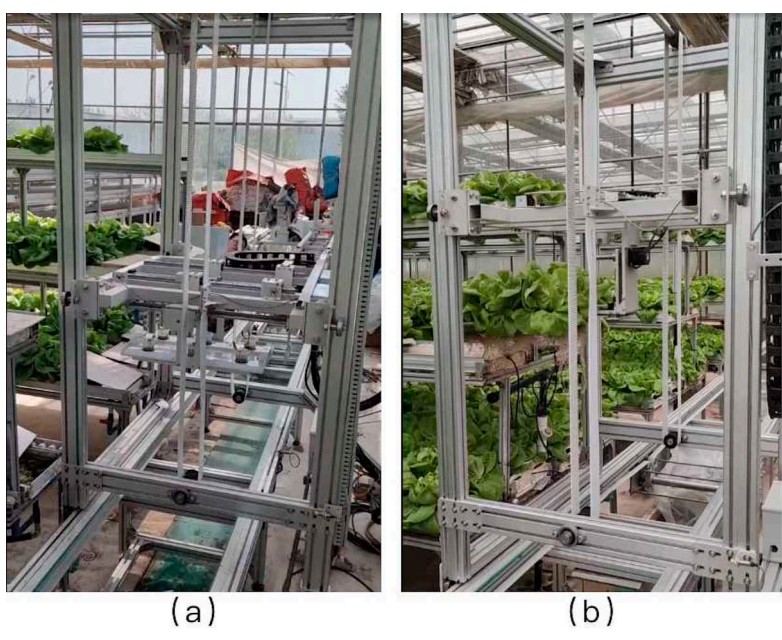

**Figure 16.** Success rate experiment site for the transport system. (**a**) Cultivation plate input experiment site; (**b**) cultivation plate output experiment site.

## 3. Results

### 3.1. Simulation Analysis of Planting Cup Propulsion Process Based on Dynamic Simulation

From the above theoretical analysis, it can be seen that the propulsion speed is a key factor affecting the stability of the planting cups during the propulsion process. To investigate the influence of propulsion speed on the stability of the planting cups, this study assumed that during the primary propulsion process, the velocity of the cultivation plate drops to zero at the moment of collision, and then a planting cup collides with the cultivation plate with a preset velocity in a very short time. In the study, we used Solid-Works to model the cultivation plate propulsion process, which includes the cultivation plate, the planting cup, and the support ground. The model was imported into ADMAS (version 2020) with the constraint types for each component as shown in Table 2 below. ADAMS 2020 software, the professional multi-body dynamic simulation software, was used because it can analyze interactions between objects [35] and therefore simulate the collision process between the planting cup and the cultivation plate. Since the maximum propulsion velocity was 0.5 m·s$^{-1}$, the simulated velocities were set to 0.5 m·s$^{-1}$, 0.4 m·s$^{-1}$, 0.3 m·s$^{-1}$, and 0.2 m·s$^{-1}$. Given that the motor can reach a stable speed in about 0.3 s, the number of simulation steps was set to 100 and the total duration was set to 1 s. The changes in the center of mass of the planting cup at different propulsion speeds during the simulation are shown in Figure 17.

**Table 2.** ADAMS parameter settings.

| Constraints | Objects |
| --- | --- |
| Moving Sub | Cultivation plate with support ground |
| | Planting cup with support ground |
| Contact | Planting cup with cultivation plate |
| | Cultivation plate with support ground |
| Drive | Cultivation plate |

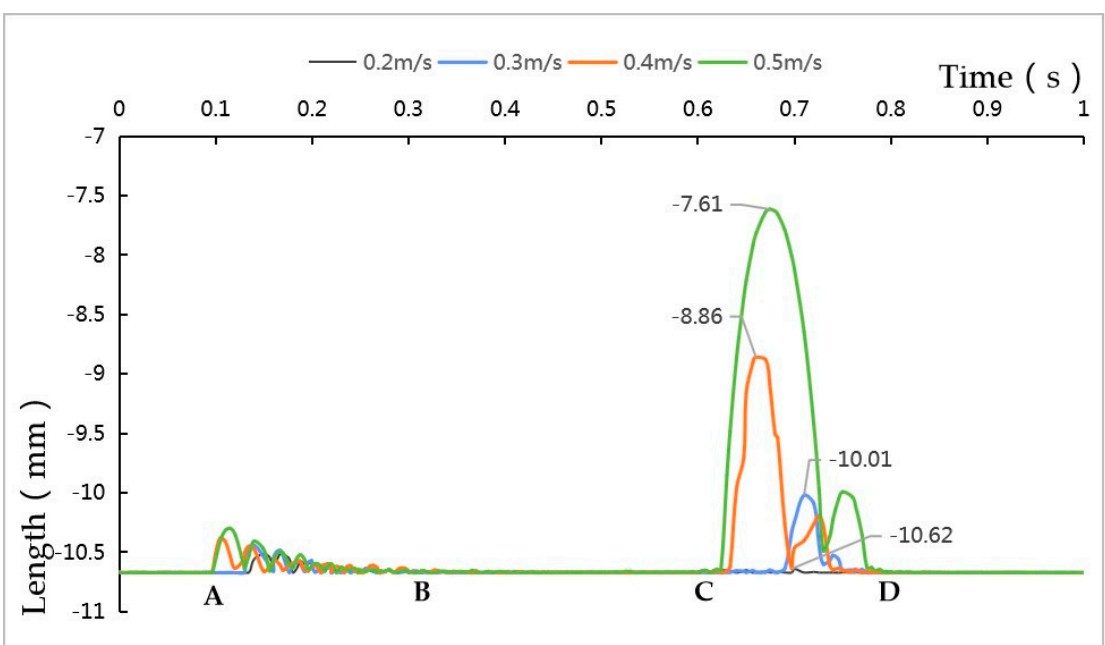

**Figure 17.** Variation curve of the center of mass displacement of planting cups. The A–B section represents the collision process between the cultivation plate and the planting cup during the startup process, while the C–D section represents the process of the planting cup continuing to collide with the cultivation plate at the preset pushing speed after the cultivation plate stops moving.

From the analysis of Figure 17, it can be seen that the maximum height changes of the center of mass were 0.04 mm, 0.65 mm, 1.80 mm, and 3.05 mm when the propulsion speed was increased from 0.2 m·s$^{-1}$ to 0.5 m·s$^{-1}$. This indicates that with the increase of the propulsion speed, the wobble of the planting cup during the propulsion process gradually increases. The initial height position of the center of mass of the planting cup was −10.66 mm. It was found that the change in the height of the center of mass was equal to 5.90 mm when the planting cup was tipped over during the simulations in SolidWorks 2020 software. The maximum changes of the center of mass of the planting cup under the four propelling speeds, accounting for the change in center of mass height at the time of tipping over of the cup, were 0.68%, 11.02%, 30.51%, and 51.69%, respectively. Although the four propulsion speeds did not reach the condition of cup overturning, considering the stability of the propulsion process, the propulsion speed of 0.3 m·s$^{-1}$ was selected in this study.

### 3.2. Static Analyses during the Output of the Cultivation Plate

To analyze the deformation and stress distribution of the cultivation plate during the output process, we conducted finite element analysis using ANSYS Workbench 2021, and a model of the cultivation board was imported into ANSYS in Parasolid format for analysis. Finite element analysis is a powerful numerical method used to simulate the behavior of complex structures under various loading conditions. In our study, we imported the model of the cultivation plate with several key parameters considered to ensure the accuracy and

reliability of the results. The material of the cultivation plate was set as ABS plastic. We applied a gravitational force of 2.79 N under each planting cup and 6.71 N in the center of the cultivation plate. To ensure the accuracy of the simulation's calculations, this study used the Sweep function to partition the mesh and set the free form surface mesh type to Quad/Tri [36]. The mesh size at the handle was set to 1 mm and the other parts were set to 3 mm, with a total of 395,322 nodes and 203,811 meshes. The deformation analysis diagram and equivalent stress cloud diagram of the cultivation plate in the vertical direction during the output process are shown in Figure 18 below.

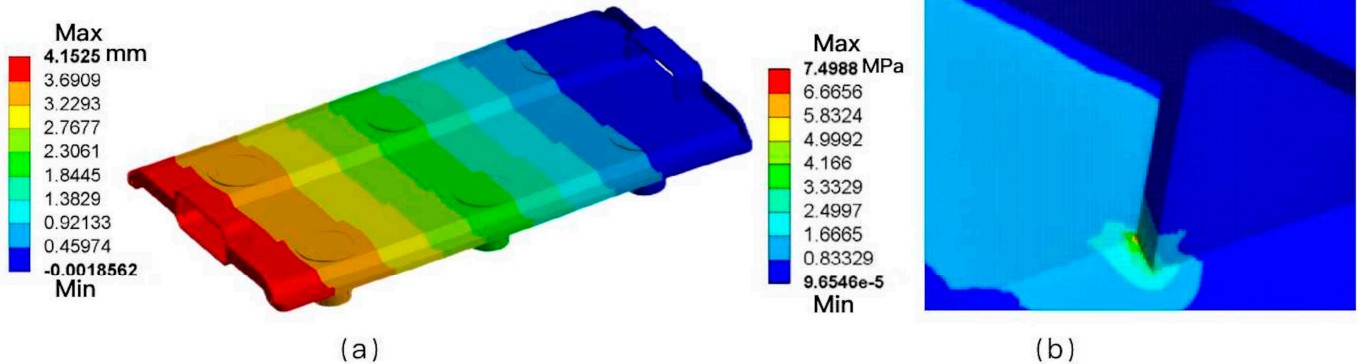

**Figure 18.** Static analysis of the cultivation plate. (**a**) Shows the distribution of deformation of the cultivation plate in the vertical direction; (**b**) shows the distribution of the equivalent stress cloud at the handle of the cultivation plate.

From Figure 18, it can be seen that the maximum deformation ($L_1$) in the vertical direction during the output of the cultivation plate was 4.15 mm, and the length ($L_0$) of the cultivation plate was 530 mm. The oblique angle of deformation in the vertical direction $\theta$ was calculated as follows:

$$\theta = arc \sin \frac{L_1}{L_0} \tag{13}$$

Through calculation, the value of $\theta$ was 0.45°, indicating the deformation of the cultivation plate was tiny, which meets its output operational requirements. From Figure 18b, it can be observed that the stress in the connection part of the handle of the cultivation plate was the largest, and the simulated value was about 7.5 MPa, while the material of the cultivation plate was ABS plastic, and the tensile yield strength of the cultivation plate was 36.3 MPa. Therefore, the gripping method of the cultivation plate will not cause any damage to it.

*3.3. Analysis of Positioning Accuracy Results*

The measurement results of the positioning error of the horizontal and vertical motion mechanisms are shown in Table 3 at both horizontal and vertical running speeds of 0.1 m·s$^{-1}$, 0.2 m·s$^{-1}$, 0.3 m·s$^{-1}$, and 0.4 m·s$^{-1}$.

Based on the results of the data in Table 3, it can be seen that the average positioning error of the transport system in the horizontal direction (X-axis direction) increases with an increase in running speed. This phenomenon may stem from the fact that the overall weight of the equipment is large, and the increase in running speed causes the inertia of the equipment itself to increase the positioning error. The maximum positioning error in the horizontal direction was 14.33 mm when the running speed reached 0.4 m·s$^{-1}$. The maximum positioning error in the horizontal direction was 8.42 mm when the running speed was 0.3 m·s$^{-1}$, and the allowable amount of positioning error for the output actuator and the input actuator in the horizontal direction is 7.5 mm (shown in Figure 11) and 8 mm (shown in Figure 19a). Therefore, under the condition of taking into account the transport efficiency and the allowable amount of positioning error, the running speed in the

horizontal direction that should be selected is 0.2 m·s$^{-1}$; as such, the maximum positioning error is 2.87 mm and the relative error is 0.8%.

**Table 3.** Horizontal and vertical positioning error measurement results.

| Running Speed (m·s$^{-1}$) | Horizontal Direction | | | Vertical Direction | | |
|---|---|---|---|---|---|---|
| | Horizontal Theoretical Displacement (mm) | Horizontal Positioning Error (mm) | Horizontal Relative Error (%) | Vertical Theoretical Displacement (mm) | Vertical Positioning Error (mm) | Vertical Relative Error (%) |
| 0.1 | 180 | 0.87 | 0.48% | 114 | 0.86 | 0.75% |
| | 359 | 1.30 | 0.36% | 799 | 0.72 | 0.09% |
| | 718 | 1.13 | 0.16% | 1484 | 0.84 | 0.06% |
| | Average value | 1.10 | | | 0.81 | |
| 0.2 | 180 | 2.27 | 1.26% | 114 | 0.80 | 0.70% |
| | 359 | 2.87 | 0.80% | 799 | 0.93 | 0.12% |
| | 718 | 2.13 | 0.30% | 1484 | 0.97 | 0.07% |
| | Average value | 2.42 | | | 0.90 | |
| 0.3 | 180 | 7.35 | 4.08% | 114 | 1.12 | 0.98% |
| | 359 | 8.42 | 2.35% | 799 | 0.98 | 0.12% |
| | 718 | 7.95 | 1.11% | 1484 | 1.34 | 0.09% |
| | Average value | 7.91 | | | 1.15 | |
| 0.4 | 180 | 13.23 | 7.35% | 114 | 4.73 | 4.15% |
| | 359 | 14.33 | 3.99% | 799 | 5.36 | 0.67% |
| | 718 | 14.20 | 1.98% | 1484 | 5.10 | 0.34% |
| | Average value | 13.92 | | | 5.06 | |

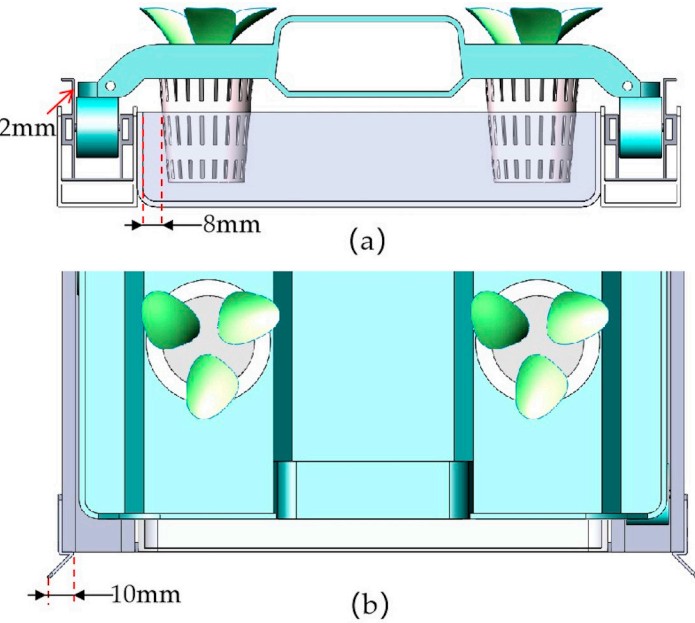

**Figure 19.** Schematic diagram of the allowable amount of positioning error in the horizontal direction of the input end of the cultivation plate. (**a**) is a schematic diagram of the allowable amount of positioning error in the horizontal direction of the cultivation trough; (**b**) is a schematic diagram of the mounting of the guide plate.

In order to make the cultivation plate more stable when it is pushed in the cultivation trough, the spacing between the cultivation plate and the slide was set to 2 mm (Figure 19a), which also lead to a smaller allowable amount of positioning error of the cultivation plate

in the horizontal direction. To solve this problem, this study set a guide plate at the input end of the cultivation trough (Figure 19b). To increase the allowable amount of positioning error in the horizontal direction through the guiding effect of the guide plate, without affecting the neighboring components in the cultivation trough installation, the guide plate's horizontal width was set to 10 mm, and the angle with the end face of the cultivation trough was set to 45 degrees.

The results of the positioning error in the vertical direction (Y-axis direction) in Table 3 show that the average positioning error in the vertical direction tended to increase with an increase of the running speed; however, when the speed was not more than 0.3 m·s$^{-1}$, the difference in the average positioning error in the vertical direction was tiny. When the speed was 0.4 m·s$^{-1}$, the average positioning error suddenly increased. During the experiments, it was found that this was due to the increase in speed in the vertical direction, which led to a slight tilt of the lift platform due to inertia at the time of stopping, thus increasing the positioning error in the vertical direction. This tilt may also lead to the failure of the gripping process. Therefore, based on positioning accuracy and system stability, the running speed in the vertical direction was selected to be 0.3 m·s$^{-1}$, which produced a maximum positioning error of 1.34 mm. During the input actuator's operation, the bottom of the planting cups was set to be higher than the upper end of the cultivation trough by 5 mm to eliminate the positioning error in the vertical direction. During the output process, the allowable amount of positioning error in the vertical direction is 5 mm, which can accommodate these positioning errors in the vertical direction.

### 3.4. The Success Rate of the Transport System

The results of the experiments on the success rate of each layer of the transport system under the conditions of 0.2 m·s$^{-1}$ for the horizontal motion mechanism and 0.3 m·s$^{-1}$ for the vertical motion mechanism are shown in Table 4 below.

**Table 4.** Success rate experiment results of the transport system.

| Cultivation Layer | Number of Experiments | Success Rate | Total Time | Average Time | Working Efficiency |
|---|---|---|---|---|---|
| The upper layer | 50 | 96.0% | 1025 s | 20.50 s | 176 plates/h |
| The middle layer | 100 | 94.0% | 1574 s | 15.74 s | 229 plates/h |
| The lower layer | 200 | 92.5% | 2267 s | 11.34 s | 317 plates/h |
| Average value | | 94.2% | | | 241 plates/h |

According to the data in Table 4, we can see that the upper layer transport success rate reached 96.0%, the lower layer transport success rate was 92.5%, and the average transport success rate was 94.2%. The working efficiency of the transport system ranged from 176 plates/h to 317 plates/h, and the average working efficiency was 241 plates/h. These experiments confirmed that the transport system had a high success rate and working efficiency, and could meet the requirements of cultivation plate transport equipment in stereoscopic cultivation, and also provided a technical guarantee for the automation of the production of the stereoscopic cultivation system.

The main reasons for the failures during the experiments were as follows: Firstly, with an increase in the number of transport times, the vertical direction error gradually accumulated, which in turn lead to the failure of the output process. Secondly, there was a positional deviation in some of the cultivation plates transported from the ground conveyor line, which interfered with the guiding plate in the propulsion process and thus lead to the failure of the input. Thirdly, the some cultivation plates were not lifted enough during the output process, which lead to part of the plant root system interfering with the cultivation trough, which in turn triggered the cultivation plate to deviate and it could not be placed on the ground conveyor line. In order to solve the above problems, measures such as optimizing the control method, adding a restraining mechanism at the supply side of the cultivation plate, and optimal operating parameters can be taken. As far as the transport of cultivation plates is concerned, the research has the ability to meet the demand.

## 4. Discussion

(1) In this study, automatic cultivation plate transport equipment was designed for stereoscopic cultivation in plant factories which can realize the automatic input and output of cultivation plates. Through in-depth research on the structure and transport methods of the cultivation plate, the study systematically designed the lift cart, input actuator, and output actuator, and the structural design and parameter calculation of the key components of each mechanism were carried out. We also simulated and verified the key operation processes of the transport system through dynamic simulation and finite element numerical simulation technology. The transport success rate experiments showed that the productivity and success rate of the transport system for handling cultivation plates are high. The manual mode of cultivation plate transportation is the reciprocating transportation, and its transportation principle is similar to that of AGV carts. In contrast, this transport system adopts the progressive transport type, and the transportation efficiency is 115–250% higher than the reciprocating transportation. Under the experimental conditions of stereoscopic cultivation, the average input-output efficiency of the transportation system was 241 plates/hour, which is much higher than the estimated average efficiency of manual operation (approximately 71 plates/h), reaching about 3.4 times of the efficiency of manual operation. Furthermore, the system does not have the fatigue and durability issues that are commonly found in manual operation. Therefore, the cultivation plate transport system designed in this study demonstrated an effective and stable transport performance, and it is of practical application significance for upgrading the automation level of stereoscopic cultivation in plant factories.

(2) This study found that positioning accuracy is one of the most important factors that affect the automation level and performance of the transport system, and the installing error rate of the stereoscopic cultivation frame should be controlled within a reasonable range of positioning error, indicating that it is an important direction to further improve the automation level and success rate of this equipment. This study adopted high-precision transmission design and servo control technology to limit the maximum positioning error in the horizontal direction to 2.87 mm and the maximum positioning error in the vertical direction to 1.34 mm, and these positioning accuracies are much better than those of the scissor-type multifunctional operation platforms developed by Yu and colleagues [22], whose maximum vertical positioning error was about 9 mm, and also better than that of the cultivation plate transport equipment designed by Zhou and colleagues [23], whose average positioning error at the same guide speed was about 10 mm. Therefore, the positioning accuracy of the transport system designed in this study is higher than that of the more common shear fork transport systems, and its accuracy can be further improved with the standardization and precision of the installation processes of stereoscopic cultivation frames.

(3) In addition to improving productivity and automation, this cultivation plate handling equipment has been optimized in terms of applicability and cost. Considering the complexity and variability of cultivation situations in various plant factories, this system can assist in the distribution of cultivation plates by adding multiple handling systems to the conveyor line, modifying the plate supply scheme, and adding mechanical structures. Especially when the cultivation frames are high, the transport system can be designed with various specifications and models by changing the external dimensions of the lift carts and limit rails, etc., without changing the basic working principles, so as to adapt to the transport operational requirements of plant factories of different scales. Meanwhile, all the key subsystems in this transport system adopt standard components in the market without special customization, as this can make its installation cost lower than that of most logistics handling equipment and enhance its promotional value. The cost advantage of this transport system can be expanded in larger-scale plant factories of various types to achieve a higher return-on-investment ratio, and also provides a technical guarantee for the automated operation of stereoscopic cultivation systems.

(4) One limitation of this study is that the horizontal motion of the lift cart adopts a unilateral power supply design, which avoids the problem of interference between the bilateral power transmission shaft and the ground conveyor line legs while maintaining a compact structure, but also leads to uneven force on the overall frame, which is prone to shaking when the running speed increases. In the future, we will attempt to adjust the unilateral power to a bilateral power mode, which will further improve the stability and productivity of its operation. The transport system should be optimized regarding operational stability and mode of operation to adapt to plant factories of different scales, so as to further improve the efficiency of plant factory transport operations and reduce production costs. In addition, it is inevitable that the future development of plant factories will be in the direction of low energy consumption and high efficiency. As such, researchers should continuously explore the integration and development of automated equipment and efficient cultivation modes in plant factories, and also explore the introduction of robotics and artificial intelligence technology, improving the level of intelligence and operational efficiency of plant factories.

## 5. Conclusions

1.  In plant factory stereoscopic cultivation systems, the efficiency of the cultivation plate transport processes is affected by problems such as high labor intensity, low levels of automation, and poor versatility of existing solutions. Therefore, the present study designed a cultivation plate transport system that can automatically input and output cultivation plates, and can flexibly adjust its structure to accommodate different cultivation frame heights. Through in-depth research on the structure and transport methods of cultivation plates, this study systematically designed a lift cart, input actuator, and output actuator, and the structural design and parameter calculation of the key components of each mechanism was carried out.

2.  During the input process of the cultivation plates, the force analysis of the planting cups showed that the propulsion speed was the main factor affecting their overturning. The center-of-mass displacement curve of a planting cup under different propulsion speeds was simulated using ADAMS 2020 software, and the optimal propulsion speed was determined to be 0.3 m·s$^{-1}$. During the output process, the ANSYS was used to analyze the static mechanics of cultivation plates, and the results showed that the maximum deformation of a cultivation plate in the vertical direction was 4.15 mm, the tilt angle was 0.45°, and the maximum stress at the handle was 7.50 MPa, which was lower than the yield strength of the cultivation plate. This indicates that the output method of the cultivation plates meets the design requirements and does not cause damage.

3.  In this study, operation performance experiments were conducted for the transport system, and the experiments showed that the average positioning error in the horizontal direction gradually increased with the increase of the operating speed, while the magnitude change in the vertical direction was small and not significant. Under the premise of ensuring the allowable amount of positioning error in the horizontal and vertical directions, when the running speed of the horizontal motion was 0.2 m·s$^{-1}$, the maximum positioning error was 2.87 mm, and the relative error was 0.80%. When the running speed of the vertical motion mechanism was 0.3 m·s$^{-1}$, the maximum positioning error generated was 1.34 mm and the relative error was 0.09%. Under the above two speed conditions, the transport success rates of each layer of the cultivation plate were further tested, and the results showed that the transport success rates of each layer were between 92.5% and 96.0%, and the operation performance could reach from 176 plates/h to 317 plates/h. The experiments confirmed that the cultivation plate transport system designed in this study demonstrated a stable transport performance, and it is of practical application significance for upgrading the automation level of stereoscopic cultivation in plant factories.

**Supplementary Materials:** The following supporting information can be downloaded at: https://www.mdpi.com/article/10.3390/agriculture14030488/s1, Figure S1. The control flowchart of the input system. Figure S2. The control flowchart of the output system. Figure S3. The circuit diagram of the input system. Figure S4. The circuit diagram of the output system. Table S1. Electrical component parameter table.

**Author Contributions:** Conceptualization, D.J. and G.G.; methodology, D.J.; software, D.J.; validation, W.G. and L.W.; formal analysis, D.J.; investigation, D.J. and W.Z.; resources, W.Z.; data curation, D.J.; writing—original draft preparation, D.J.; writing—review and editing, G.G.; visualization, L.W.; supervision, G.G.; project administration, W.G.; funding acquisition, W.Z. All authors have read and agreed to the published version of the manuscript.

**Funding:** This research was funded by the Ningxia Hui Autonomous Region Key Research and Development Program (2023BCF01047).

**Institutional Review Board Statement:** Not applicable.

**Data Availability Statement:** Data are contained within the article and Supplementary Material.

**Conflicts of Interest:** The authors declare no conflicts of interest.

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
