# Peer review of "Design and Experiment of Automatic Transport System for Planting Plate in Plant Factory"

_agriculture, doi:10.3390/agriculture14030488_

Round 1

Reviewer 1 Report

Comments and Suggestions for Authors

1. This paper designs an automatic cultivation plate transportation equipment, which is a general equipment without innovation.

2, the article is more like a course design, lack of academic.

Comments on the Quality of English Language

Moderate editing of English language required

Reviewer 2 Report

Comments and Suggestions for Authors

This paper focuses on the automatic design of the tray transportation system in the stereoscopic culture system of the plant factory. The purpose of the research is to improve the automation degree of the plant factory, reduce the labor intensity, and solve the problem that the existing transportation equipment can not adapt to the customization of different three-dimensional culture modes. The research has practical significance to improve the automation level and mechanization of plant factories, and is helpful to develop three-dimensional culture mode and increase output. However, there are still some suggestions for your reference before publication:

1. In the bottom of Introduction, please provide the simple descriptions of Sections 2-5.

2. Considering that the specific application of motor control technology in smart agriculture should be briefly mentioned in the background introduction, as the basis for comprehensive consideration of the background of the article, the following articles may help you:

(1)Design and Experiment of a Targeted Variable Fertilization Control System for Deep Application of Liquid Fertilizer. Agronomy 2023, 13, 1687.

(2)GAPSO-Optimized Fuzzy PID Controller for Electric-Driven Seeding. Sensors 2022, 22, 6678.

3. Figure 18 Does the vertical and horizontal axes have units? Mark it if you do.

4. All variables in the article should be italicized. Please check the whole article and correct it.

5. Please give the technical key parameters of the motors and sensors used, so that other scholars can carry out similar studies in the future, and suggest making a table

6. Please give the control flow chart and the key part of the circuit diagram to complete the operation logic of the control system.

7. Please give the latitude and longitude coordinates of the site of the experiment, so that readers can determine the feasibility of the test conditions.

8. Please discuss the analysis or test analysis whether the mechanical compared with manual operation in the economic cost advantage? If yes, can you explain the specific economic cost savings?

Comments on the Quality of English Language

none

Reviewer 3 Report

Comments and Suggestions for Authors

The manuscript is interesting and clear. It describes a new technology for plant factories. This innovation was projected, realized and tested. In my opinion this manuscript deserves publication after minor revision. Please find below some specific suggestions.

- Please check the keywords, they should not repeat the words of the title.

- I appreciate that this very technical manuscript fits a standard article template for Agriculture (Introduction, Materials and Methods, Results, Discussion, Conclusions) but the Materials and Methods section is too long. However removing information would affect the quality of the manuscript. Might it be possible to move some technical specifications from Materials and Methods section to Results?

- I would prefer the use of m s-1 instead of m/s

- The discussion is very clear and describes the main findings of this research but a comparison with the current literature is completely missing, please add.

- In the conclusion section I would add some information about the future perspectives in the research on Plant factories

Good luck for the publication of your manuscript and thank you in advance for taking into consideration my suggestions

Reviewer 4 Report

Comments and Suggestions for Authors

Dear Authors,

The authors have designed an automatic transport system for cultivation plates in plant factories to improve efficiency and automation. The study includes detailed descriptions of the design process, dynamic and numerical simulations, and operational performance experiments. The results indicate that the transport system meets the operational requirements and provides feasible solutions for the automation of plant factory transport equipment.The authors have addressed an important issue in plant factory cultivation, specifically the transportation of cultivation plates. The design of an automatic transport system is a significant contribution to improving the efficiency and automation of plant factories. The study provides a detailed description of the design process and experimental results, which adds credibility to the findings.The use of dynamic simulation technology in determining the optimum propulsion speed for the input process is commendable. It would be helpful if the authors could provide more information on the methodology and parameters used in the dynamic simulation.The finite element numerical simulation for analyzing the deformation and stress distribution of the cultivation plate in the output process is a valuable approach. It would be beneficial to include more details about the specific numerical simulation technique employed and the validation of the simulation results.The operational performance experiments demonstrate promising results, with the transport system meeting the operational requirements and exhibiting high success rates and efficiency. However, it would be useful to discuss any limitations or challenges encountered during the experiments and potential areas for improvement.The introduction provides a good overview of the global concern regarding the decline of arable land and the need for innovative solutions in food production. It would be beneficial to include a brief discussion on the existing transportation methods in plant factories and how the proposed automatic transport system improves upon them.The challenges of the existing cultivation plate transport systems in plant factories are well-addressed. However, it would be valuable to elaborate on the specific issues related to safety hazards, alignment accuracy, and transport efficiency in the context of manual and scissor lift-based transport methods. Questions to Authors:

Can you elaborate on the specific numerical simulation technique used for analyzing the deformation and stress distribution of the cultivation plate in the output process? How were the simulation results validated? Were there any limitations or challenges encountered during the operational performance experiments? Are there any areas for improvement in the current design of the transport system? Can you discuss the existing transportation methods in plant factories and how the proposed automatic transport system improves upon them? Specifically, how does it address the issues of safety hazards, alignment accuracy, and transport efficiency associated with manual and scissor lift-based methods? Have you considered the scalability and applicability of the transport system to different types of plant factories? How adaptable is the system to accommodate variations in cultivation frame heights and other factors?Are there any plans for further research or development of the transport system? Are there any specific areas that you believe require further investigation or improvement?How does the cost-effectiveness of the proposed automatic transport system compare to manual or other existing transport methods? Have you conducted any economic analysis or feasibility studies in this regard?

In the discussion section, it would be beneficial for the authors to provide a more comprehensive comparison of their findings with existing studies in the field. By comparing their results with previous research, they can highlight the novelty and significance of their work and further validate their findings.

Specifically, the authors can compare their automatic transport system with other transportation methods used in plant factories, such as manual transport using ladders or scissor lift carts. They can discuss how their system addresses the limitations and challenges associated with these traditional methods, such as high labor intensity, low automation levels, and poor versatility.

Furthermore, the authors can compare their design and experimental results with similar studies that have developed automatic transport systems for cultivation plates in plant factories. They can discuss the similarities and differences in design approaches, system performance, and operational efficiency. This comparison can provide insights into the effectiveness and competitiveness of their proposed system.Moreover, it would be valuable for the authors to discuss the potential advantages and disadvantages of their transport system compared to other emerging technologies or approaches in the field. For example, they can explore the feasibility and benefits of incorporating robotics or artificial intelligence techniques in the transport system to enhance its automation and adaptability.

Comments on the Quality of English Language

Minor editing of English language required

Round 2

Reviewer 1 Report

Comments and Suggestions for Authors

The author has modified the article and added some theoretical contents, which has certain scientific value.

Comments on the Quality of English Language

no